# The DNA methylome of human sperm is distinct from blood with little evidence for tissue-consistent obesity associations

**Fredrika Åsenius**[1], **Tyler J. Gorrie-Stone**[2], **Ama Brew**[3], **Yasmin Panchbhaya**[4], **Elizabeth Williamson**[5], **Leonard C. Schalkwyk**[2], **Vardhman K. Rakyan**[3], **Michelle L. Holland**[6], **Sarah J. Marzi**[7,8‡*], **David J. Williams**[1‡]

1 UCL EGA Institute for Women's Health, University College London, London, United Kingdom, 2 School of Biological Sciences, University of Essex, Colchester, United Kingdom, 3 The Blizard Institute, Queen Mary University of London, London, United Kingdom, 4 UCL Genomics, Great Ormond Street Institute of Child Health, London, United Kingdom, 5 Fertility & reproductive medicine laboratory, University College Hospital, London, United Kingdom, 6 Department of Medical and Molecular Genetics, School of Basic and Medical Biosciences, King's College London, London, United Kingdom, 7 UK Dementia Research Institute, Imperial College London, London, United Kingdom, 8 Department of Brain Sciences, Imperial College London, London, United Kingdom

‡ These authors are joint senior authors on this work.
* s.marzi@imperial.ac.uk

**Citation:** Åsenius F, Gorrie-Stone TJ, Brew A, Panchbhaya Y, Williamson E, Schalkwyk LC, et al. (2020) The DNA methylome of human sperm is distinct from blood with little evidence for tissue-consistent obesity associations. PLoS Genet 16(10): e1009035. https://doi.org/10.1371/journal.pgen.1009035

**Data Availability Statement:** The data underlying the results presented in the study are available from GEO under accession number GSE149318:

## Abstract

Epidemiological research suggests that paternal obesity may increase the risk of fathering small for gestational age offspring. Studies in non-human mammals indicate that such associations could be mediated by DNA methylation changes in spermatozoa that influence offspring development in utero. Human obesity is associated with differential DNA methylation in peripheral blood. It is unclear, however, whether this differential DNA methylation is reflected in spermatozoa. We profiled genome-wide DNA methylation using the Illumina MethylationEPIC array in a cross-sectional study of matched human blood and sperm from lean (discovery n = 47; replication n = 21) and obese (n = 22) males to analyse tissue covariation of DNA methylation, and identify obesity-associated methylomic signatures. We found that DNA methylation signatures of human blood and spermatozoa are highly discordant, and methylation levels are correlated at only a minority of CpG sites (~1%). At the majority of these sites, DNA methylation appears to be influenced by genetic variation. Obesity-associated DNA methylation in blood was not generally reflected in spermatozoa, and obesity was not associated with altered covariation patterns or accelerated epigenetic ageing in the two tissues. However, one cross-tissue obesity-specific hypermethylated site (cg19357369; chr4:2429884; $P = 8.95 \times 10^{-8}$; 2% DNA methylation difference) was identified, warranting replication and further investigation. When compared to a wide range of human somatic tissue samples (n = 5,917), spermatozoa displayed differential DNA methylation across pathways enriched in transcriptional regulation. Overall, human sperm displays a unique DNA methylation profile that is highly discordant to, and practically uncorrelated with, that of matched peripheral blood. We observed that obesity was only nominally associated with differential DNA methylation in

https://www.ncbi.nlm.nih.gov/geo/query/acc.cgi?acc=GSE149318. Analysis code is publicly available on GitHub: https://github.com/SarahMarzi/BloodSperm_DNAMethylation.

**Funding:** FA was supported by a studentship from the Rosetrees Trust (Ref No A815; https://www.rosetreestrust.co.uk) and the work was supported by a Medical Research Council grant (MRC reference code MR/P011799/1; https://mrc.ukri.org). SJM is funded by the Edmond and Lily Safra Early Career Fellowship Program (https://www.edmondjsafra.org) and the UK Dementia Research Institute (https://ukdri.ac.uk), which receives its funding from UK DRI Ltd, funded by the UK Medical Research Council (https://mrc.ukri.org), Alzheimer's Society (https://www.alzheimers.org.uk) and Alzheimer's Research UK (https://www.alzheimersresearchuk.org). DJW is supported by the National Institute for Health Research University College London Hospitals Biomedical Research Centre (https://www.uclhospitals.brc.nihr.ac.uk/content/biomedical-research-centre). The funders had no role in study design, data collection and analysis, decision to publish, or preparation of the manuscript.

**Competing interests:** The authors have declared that no competing interests exist.

sperm, and therefore suggest that spermatozoal DNA methylation is an unlikely mediator of intergenerational effects of metabolic traits.

## Author summary

Research primarily conducted in mice suggests that obesity in fathers can have effects on the health of their offspring via changes in the fathers' sperm. It is not confirmed whether this is true for humans. In this study, we examined sperm and blood from lean and obese men to understand whether obesity affects DNA methylation in both tissues. DNA methylation can impact on gene function and therefore may affect offspring health. We found that there was almost no association between obesity and DNA methylation in sperm. We also showed that DNA methylation patterns found in the blood of obese individuals are not present in sperm from obese men. Generally, DNA methylation patterns across the whole genome were completely different and uncorrelated between the two tissues. Lastly, we compared DNA methylation patterns in sperm to those in many other tissues, including for example blood and brain samples, and found that sperm has a unique signature of DNA methylation—one that points to genes involved in regulating overall levels of transcription. We conclude that obesity probably does not affect DNA methylation in sperm and that, although more research is needed, if obesity in fathers does influence the health of their children, this process is unlikely to be mediated by spermatozoal DNA methylation.

## Introduction

Multiple large-scale epigenome-wide association studies in humans have shown that environmental and acquired phenotypes, including smoking, ageing and obesity, are associated with altered DNA methylation in peripheral blood [1–4]. Whether such phenotypes also have the potential to induce epigenetic changes in gametes has generated considerable interest in recent years. Studies in non-human mammals suggest that the spermatozoal DNA methylome can be influenced by factors such as dietary alterations, toxicants and even psychological stress [5–10], although the majority of these results have yet to be replicated independently. A small number of studies also suggest that acquired traits in male mice induce epigenetic changes in sperm, which in turn influence the physiology of offspring [7, 11, 12].

There is little evidence for such inter- and transgenerational effects of acquired phenotypes via epigenetic inheritance in humans. This is partly due to the fact that human sperm is rarely analysed outside of a reproductive medicine setting and is less accessible than, for example, peripheral blood. Further, it is ethically and practically impossible to perform a study of transgenerational effects in humans in which all potential external and lifestyle-related confounders are removed, and inter-individual genetic variation is generally not controllable. In addition, one needs to account for the two-stage process of epigenetic reprogramming of primordial germ cells and preimplantation embryos that occurs between generations [13]. Lastly, epigenetic signatures are highly tissue- and developmental stage specific [14, 15], making findings from studies using whole blood as a surrogate tissue for spermatozoa difficult to interpret [16].

Despite these caveats, epidemiological evidence suggests that factors such as advanced paternal age, obesity, diabetes and smoking have the potential to negatively impact the development and physiology of a man's offspring [17–19]. Such associations could be mediated by alterations to the father's spermatozoa (Fig 1A), although other possibilities include changes in

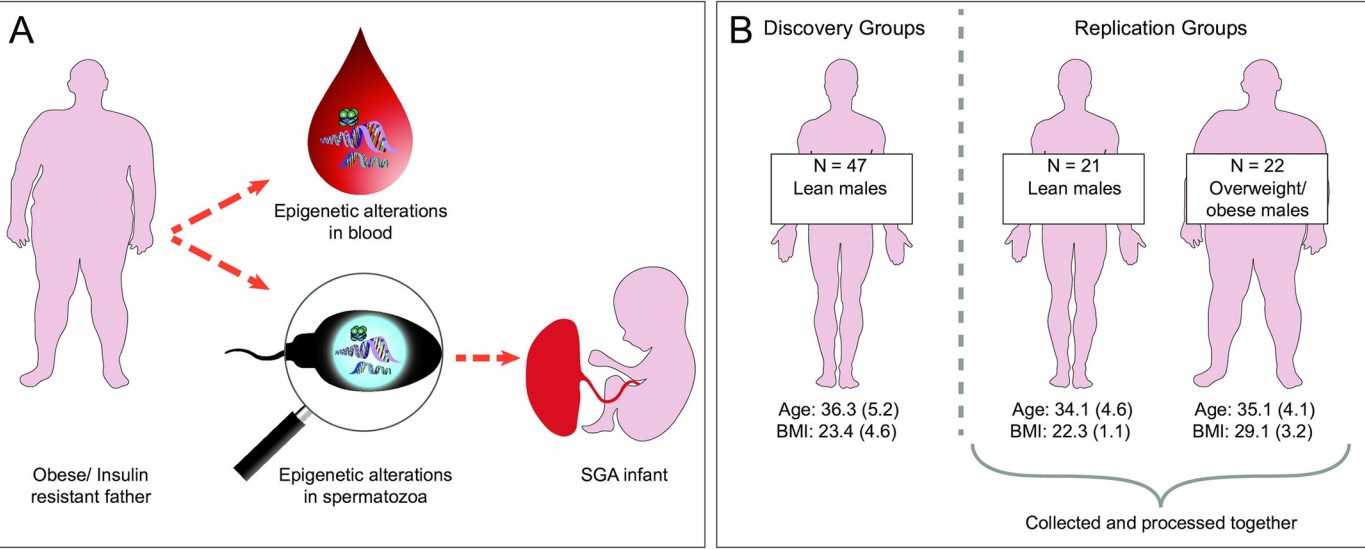

**Fig 1. Intergenerational epigenetic inheritance via spermatozoa and overview of study groups.** (**A**) Mechanism for how acquired paternal phenotypes could alter offspring physiology via epigenetic alterations to a man's spermatozoa. Epidemiological studies suggest that some acquired paternal traits, including obesity and insulin resistance, are associated with an increased risk of fathering small for gestational age (SGA) offspring [18, 19, 69]. Studies in non-human mammals suggest that such associations could be mediated by DNA methylation alterations in spermatozoa that induce metabolic reprogramming in the developing foetus [12]. (**B**) Overview of study groups. The discovery group included 47 lean males (BMI 19–25 kg/m$^2$) and the replication groups included 21 lean males (BMI 19–25 kg/m$^2$) and 22 overweight/obese males (BMI >26 kg/m$^2$; 'the obesity group'). Age (years) and BMI (kg/m$^2$) are expressed as mean (SD). *SGA: small for gestational age. SD: standard deviation.*

the composition of seminal fluid or indirect effects on the mother and, importantly, postnatal effects such as paternal behaviour. An improved understanding of whether and how acquired paternal traits can influence offspring physiology has important implications, both scientifically and in terms of public health policy. This is particularly pertinent for modifiable traits such as obesity, where timely intervention could reduce any potential negative intergenerational effects.

It will be a long time before studies of DNA methylation in human spermatozoa reach a comparable magnitude to those currently available on peripheral blood. Therefore, it is of interest to identify CpG sites where DNA methylation levels covary between the two tissues, that is, sites at which blood methylation is predictive of sperm methylation, even if the absolute level of methylation is different. The extent to which these sites overlap with those identified in blood as associated with environmental stimuli or acquired phenotypes will provide new insight into whether the sperm methylome may be similarly responsive. At such CpG sites, using blood DNA methylation as a proxy for inferring DNA methylation in spermatozoa might be justified. To our knowledge, the largest study that analysed genome-wide DNA methylation in an unbiased manner in matched samples of blood and sperm to date included a total of eight participants [20].

In this study, we analysed genome-wide DNA methylation using the Infinium MethylationEPIC array in matched samples of human blood and sperm from lean (n = 68; BMI <25kg/m$^2$) and overweight/obese (n = 22; BMI >26kg/m$^2$; 'the obesity group') healthy males of proven fertility (Fig 1B). We interrogated the extent to which obesity-associated DNA methylation in blood is reflected in spermatozoa from obese males and identified obesity associated CpG-sites in sperm and blood. Spermatozoal DNA methylation data was further compared to that of nearly 6,000 somatic tissue samples available on the Gene Expression Omnibus data repository [21], allowing us to identify sperm-specific DNA methylation signatures. Together,

our analyses interrogate the plausibility of spermatozoal DNA methylation as a mechanism for intergenerational effects of paternal obesity and whether whole blood can be used as a surrogate tissue for analyses of DNA methylation when sperm is unavailable. Further, they provide a unique insight into how spermatozoal DNA methylation compares to DNA methylation in a wide range of human somatic tissues.

# Results

## General characterisation of the sperm DNA methylome

We used the Illumina MethylationEPIC array to quantify DNA methylation at > 850,000 CpG sites across the human genome in matched samples of whole blood and sperm from a discovery group of 47 lean, healthy males of proven fertility. Following pre-processing, normalization and stringent quality control (see Materials and methods), a total of 704,356 probes were retained for further analyses. Raw and pre-processed DNA methylation data is available for download from the Gene Expression Omnibus (GEO) at accession number GSE149318. To characterize spermatozoal DNA methylation across genomic regions, levels of DNA methylation were divided into three categories; 'low', 'intermediate' and 'high', corresponding to median DNA methylation < 20%, 20–80% and > 80% across individuals respectively (Fig 2). As observed in other tissues and cell types, CpG islands and shores generally show low DNA methylation in sperm. Conversely, sites mapping to the open sea were characterized by overall higher DNA methylation (Fig 2A, S1 Table). Gene bodies in spermatozoa displayed overall high levels of DNA methylation, whilst sparser DNA methylation was seen around transcription start sites (TSS) and 5' untranslated regions (UTRs), as well as the first exons (Fig 2B, S2 Table).

## DNA methylation in imprinted regions

Genomic imprinting refers to the phenomenon that genes are epigenetically regulated to be expressed in a parent-of-origin specific manner [22]. In spermatozoa, imprinted genes should be either completely unmethylated or fully methylated depending on the gene [22]. Conversely, in blood, the parent-of-origin driven allele-specific methylation should result in methylation values of around 50% for any given imprinted site. DNA methylation levels at CpG sites annotated to genes listed in the Geneimprint database (http://www.geneimprint.com/site/genes-by-species) were compared between spermatozoa and whole blood (S1 Fig). In the case of CpG sites annotated to genes that are known to be imprinted, we observed an enrichment of sites with median DNA methylation of 50% in whole blood, particularly for paternally imprinted genes (21% sites with 40–60% median DNA methylation vs 3% of sites across the array-wide background; $P < 1.00 \times 10^{-50}$, Fisher's exact test), but also for maternally imprinted genes (11% of sites; $P = 9.19 \times 10^{-9}$). For genes predicted to be imprinted according to the Geneimprint database, there was a less pronounced enrichment (paternal: 6% of sites; $P = 0.01$; maternal: 6% of sites; $P = 0.04$). No such enrichment was observed for spermatozoal DNA methylation in any of the four categories ($P > 0.05$). Because gene annotation on the methylation array is based only on proximity, this approach includes many CpG sites not actually located in imprinting control regions (ICRs). Therefore, we also compared DNA methylation distributions at sites which specifically fall into known human ICRs as reported by WAMIDEX [23]. This second approach further confirmed an enrichment of probes with around 50% methylation located in ICRs in blood compared to sperm (S2 Fig). Strikingly, of the 169 CpG sites that fell into ICRs, the majority show median DNA methylation around 50% (57% of sites with 40–60% DNA methylation, $P < 1.00 \times 10^{-50}$, Fisher's exact test vs array-

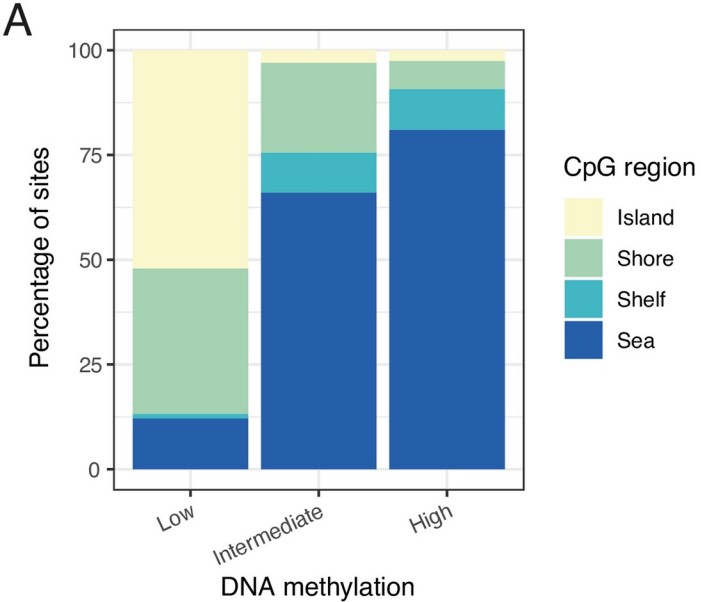

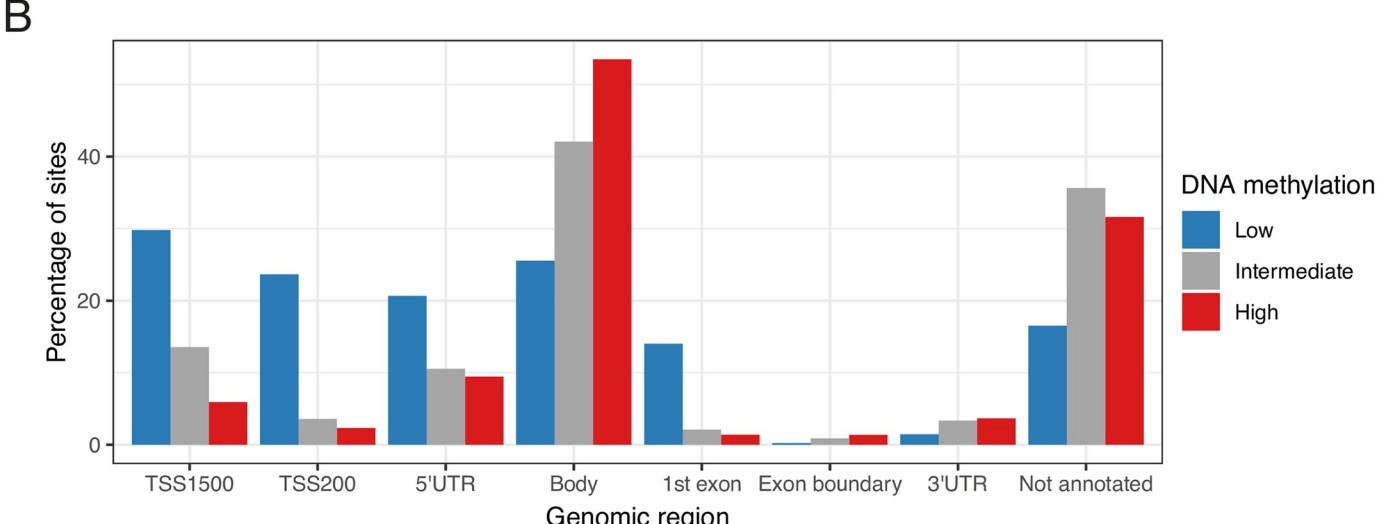

**Fig 2. DNA methylation distribution of the human sperm DNA methylome.** (**A**) The percentage of CpG sites that display low (median DNA methylation < 20%), intermediate (40–60% median DNA methylation) and high (median DNA methylation > 80%) levels of DNA methylation in spermatozoa are shown according to CpG region. (**B**) The percentage of CpG sites that display low, intermediate and high levels of DNA methylation in spermatozoa are shown according to their genomic region. *TSS*: *transcription start site*, *UTR*: *untranslated region*.

wide background). On the other hand, nearly all of the 169 sites were completely unmethylated in sperm (94% with median DNA methylation < 20%, $P < 1.00 \times 10^{-50}$).

## The sperm DNA methylome exhibits a more polarised genome-wide DNA methylation profile than blood

We compared the overall distribution of DNA methylation levels across the blood and sperm genomes. Sperm displayed a more polarised methylation profile compared to blood, i.e. that both low and high median levels of methylation were more commonly seen in sperm (Fig 3A), with 33% of sites showing median DNA methylation < 20% in sperm vs 27% in blood and

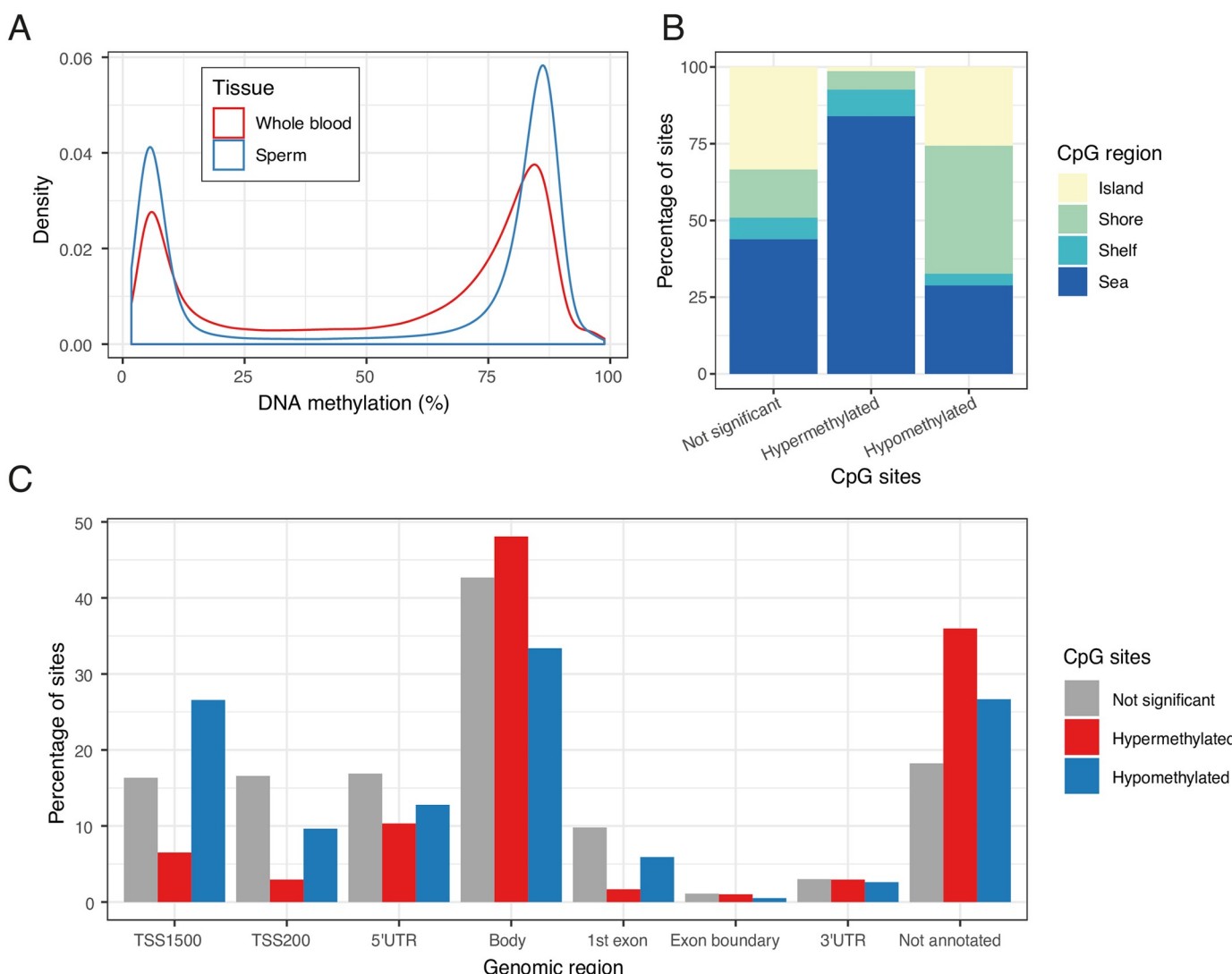

**Fig 3. Comparison of DNA methylation levels in human sperm and whole blood.** (**A**) Array-wide comparison of CpG methylation in sperm and blood, showing that both low (< 20%) and high (> 80%) DNA methylation levels are more commonly seen in sperm. Plotted is the distribution median DNA methylation levels across all individuals in the discovery group. (**B**) The percentage of CpG sites that are relatively hyper- and hypomethylated in sperm compared to blood, and CpG sites where there is no significant difference in DNA methylation between the tissues, are shown according to CpG region. (**C**) The percentage of CpG sites that are relatively hyper- and hypomethylated in sperm compared to blood, and CpG sites where there is no significant difference in DNA methylation between the tissues, are shown according to genomic region. *TSS: transcription start site*, *UTR: untranslated region*.

49% of sites with median DNA methylation > 80% in sperm vs 35% in blood. Principal component (PC) analysis was performed across the full discovery dataset comprising the 704,356 probes that remained after filtering. The first PC, explaining 51.41% of the variance, clearly distinguished between sperm and blood, indicating that the tissue of origin was the primary determinant of differences in DNA methylation profiles (S3 Fig). At the majority of interrogated sites, DNA methylation levels differed significantly between sperm and blood (n = 447,846 sites (64%), $P < 9 \times 10^{-8}$, paired t-test; S3 Table). At 62% of these sites (n = 277,831 sites), sperm was relatively hypermethylated compared to blood.

A more detailed characterisation of the differences between the sperm and blood DNA methylomes was performed by comparing DNA methylation levels in sperm and blood across different genomic regions (Fig 3, S5 and S6 Tables). CpG islands and CpG island shores were

found to be less methylated in sperm compared to blood (7% and 16% lower in sperm respectively, $P < 1.0 \times 10^{-50}$ for both, paired t-test). CpG island shelves and CpG sites in open seas were relatively hypermethylated in sperm compared to blood (6% and 7% higher in sperm respectively, $P < 1.0 \times 10^{-50}$ for both) (Fig 3B, S5 Table). Regions upstream of transcriptional start sites were relatively hypomethylated in sperm compared to blood (2% lower at TSS200 and 0.11 at TSS1500, $P < 1.0 \times 10^{-50}$ for both), as were sites mapping to the 3'UTR (1% lower, $P = 3.81 \times 10^{-5}$) or first exon (1% lower, $P < 1.0 \times 10^{-50}$). Conversely, other transcribed regions were hypermethylated in sperm compared to blood, including gene bodies (2% higher, $P < 1.0 \times 10^{-50}$), 5'UTRs (1% higher, $P = 1.3.61 \times 10^{-32}$), and exon boundaries (2% higher, $P = 2.80 \times 10^{-22}$; Fig 3C, S6 Table). We replicated these differences in the lean replication (n = 21 lean males) and obesity groups (n = 22 overweight/obese males) (S1 Text, S4 Fig and S3 Table).

## Sperm has a unique DNA methylation profile enriched in pathways relating to transcriptional regulation

The Gene Expression Omnibus (GEO) is a publicly available data repository that contains DNA methylation data from a range of human tissue samples, most of which have been analysed using the Illumina Infinium HumanMethylation450 BeadChip (450K array) [21]. In order to investigate how the DNA methylation profile of spermatozoa compares to that of somatic tissues, DNA methylation data from 371 sperm samples (90 from our discovery, replication and obesity groups combined and 281 samples from GEO) was compared to that of 5,917 somatic tissue samples from male donors available on GEO (see S7 and S8 Tables for details on tissue samples). Restricting analysis to CpG sites covered by both the EPIC and 450K arrays (n = 452,626 sites) we used linear regression to identify sperm-specific DNA methylation signals across the 6,288 samples. After Bonferroni correction, a total of 133,125 genome-wide significant CpG sites (29%) were identified as differentially methylated between sperm and somatic tissues (S9 Table). At 18% of these sites (n = 109,290 sites) sperm was characterized by higher methylation levels than somatic tissues. This is in contrast to the paired analysis with blood and likely due to the nearly exclusive coverage of CpG islands on the 450K array. Gene Ontology (GO) enrichment analysis [24] revealed 272 GO terms amongst hypermethylated CpG sites (S10 Table). The main two categories of enriched pathways related to regulation of gene transcription (37 pathways) and neurological traits and functions (67 pathways). The latter is possibly driven by the relatively large proportion of brain and neuronal samples amongst the somatic tissues (16%). Of the 37 GO terms enriched amongst hypomethylated CpG sites, 8 (22%) related to sensory perception, particularly smell (S11 Table). We repeated the same analysis removing unsorted tissues and tumours as well as cell lines (1,046 samples) and replicated virtually the same results.

## Covariation of DNA methylation between sperm and blood is limited and most likely explained by genetic variation

We next explored whether, despite the blood and sperm DNA methylomes being highly distinct, there were CpG sites where the levels of DNA methylation covaried between the tissues. We used minimum variability criteria for sites to be tested to avoid correlations driven by individual outliers, similar to those used by Hannon and colleagues [15]: we selected sites for which the middle 80% of samples had a DNA methylation range $\geq$ 5% in both blood and sperm. This restricted our analyses to 155,269 variable sites. At 1,513 of these (~1%), DNA methylation levels were significantly correlated between the two tissues ($P < 9 \times 10^{-8}$, Pearson's product moment correlation; Fig 4A, S12 Table).

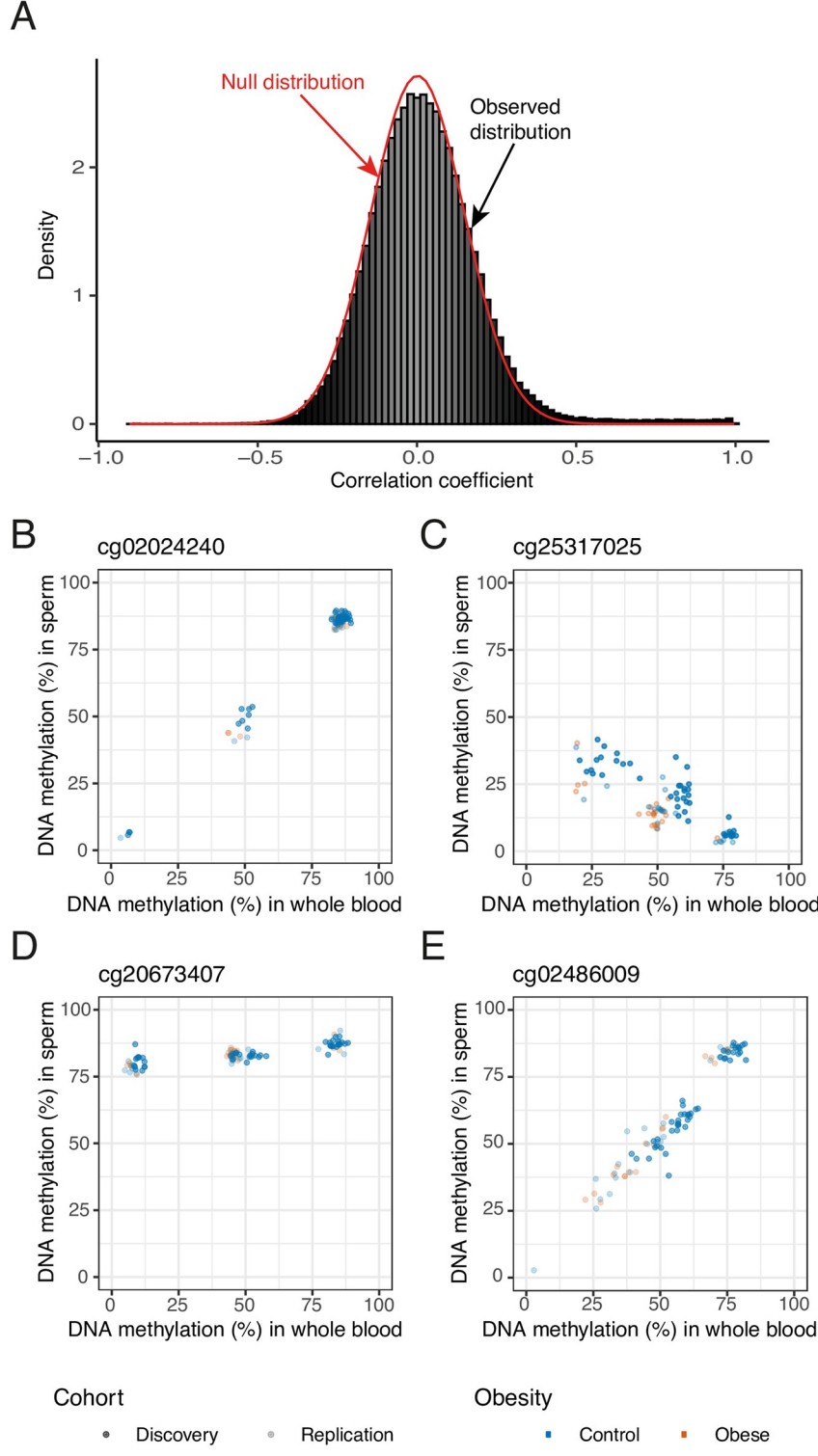

**Fig 4. Covariation of DNA methylation between blood and sperm. (A)** The observed correlation of DNA methylation levels in sperm and blood (histogram) is plotted against the estimated null distribution (red density curve). A small percentage of sites display highly correlated DNA methylation levels ($r > 0.8$), and the observed distribution is overall slightly shifted to the right compared to the null distribution. **(B)** cg02024240 (chr5:159669974) shows a strong DNA methylation correlation between blood and sperm and a trimodal methylation pattern suggestive of a genetically driven effect ($r > 0.99$, $P = 4.68 \times 10^{-48}$). **(C)** cg25317025 (chr18:47019823) is one of 30 sites showing a

negative correlation between blood and sperm ($r$ = -0.89, $P$ = $5.14 \times 10^{-17}$). (**D**) Some probes display striking differences in variability between the two tissues: cg20673407 (chr10:31040939) is characterized by a distinct trimodal pattern in whole blood while showing less overall variability in sperm ($r$ = 0.82, $P$ = $1.45 \times 10^{-12}$). (**E**) Only 6 of the significantly correlated probes have no known SNPs anywhere in the probe sequence. cg02486009 (chr15: 22428395) is one of these (r = 0.96, P = $1.90 \times 10^{-27}$). Nonetheless it shows a bimodal DNA methylation pattern in both tissues, suggestive of a genetically driven effect.

Given the observation of several bi- and trimodal patterns of DNA methylation amongst highly correlated sites (Fig 4B), we applied two separate methods (see Materials & methods), to identify which of the 1,513 significantly correlated CpG sites exhibit these patterns. The majority of correlated CpG sites showed a bimodal distribution (kmeans method: 1,140 (75%); gaphunter: 885 (58%) in blood, 898 (59%) in sperm) and a substantial number of sites were characterized by a trimodal distribution (kmeans method: 205 (14%); gaphunter: 355 (23%) in blood, 367 in sperm (24%)). These strong bi- and trimodal distributions are suggestive of a strong genetic influence on DNA methylation or the measurement thereof. Such effects could for example arise from SNPs in the CpG sites themselves (where the methylation value would represent a genotype call rather than methylation measurement), from very strong mQTLs leading to three distinct levels of methylation for the three genotypes at the QTL, or possibly because a SNP in the probe sequence is biasing the measurement of DNA methylation. Probes with the highest correlation coefficients tended to show clear trimodal patterns (Fig 4B), while a third of bimodally distributed probes appear to be driven by single outliers (kmeans method: 365 (32%); gaphunter: 369 (42%) in blood, 381 (42%) in sperm; S5 Fig). A subset of correlated sites (30 i.e. 2%) displayed a negative correlation between DNA methylation in sperm and blood (Fig 4C) and at a small number of sites distinct trimodal methylation patterns are present in only one of the two tissues (Fig 4D).

We cross-checked all correlated sites for known SNPs in the probe sequence using the dbSNP Human Build 151 database [25]. Nearly all probes (1,507; > 99%) were found to have known SNPs in the probe sequence, > 90% of which are in the CpG site itself (Fig 5). This would indicate that DNA methylation readouts at these sites are most likely measuring genetic variation rather than epigenetic state. Only a small subset (n = 6) of the CpG sites that were significantly correlated had no known SNPs in their probe sequence. Some of these nevertheless displayed bi- and trimodal patterns of DNA methylation suggestive of a genetically driven effect and could potentially constitute strong mQTLs (Fig 4E).

Secondly, we overlapped our correlated CpG sites with a list of recently reported correlated regions of systemic interindividual variation (CorSIV) in DNA methylation [26]. Only 0.2% of non-correlated variable probes are contained in CorSIVs—in line with the low overall genomic prevalence of these regions (0.1% of the human genome). Strikingly, we observe a 10-fold enrichment of this within the correlated sites (2.2%, $P$ = $8.85 \times 10^{-25}$, Fisher's exact test). The observations from the sperm data suggest that for sites exhibiting bi- and trimodal methylation patterns there is a likely genetic origin (of either a SNP in the CpG site or strong methylation QTL effects). Therefore, this enrichment conflicts with the hypothesis that for at least these sites, the origin of cross-tissue covariation is developmentally established stable epialleles [27]. Finally, using cis DNA methylation QTL data from whole blood published by McClay and colleagues [28] we found that 232 (30%) of the correlated sites also present on the 450K array had previously been identified as mQTLs in whole blood, representing a significant enrichment over the 16% observed across all variable probes ($P$ = $1.66 \times 10^{-33}$, Fisher's exact test). Correlations largely replicated in the two replication groups. (S1 Text, S12 Table) and non-replicating sites were generally driven by outliers in the discovery group (examples shown in S6 Fig).

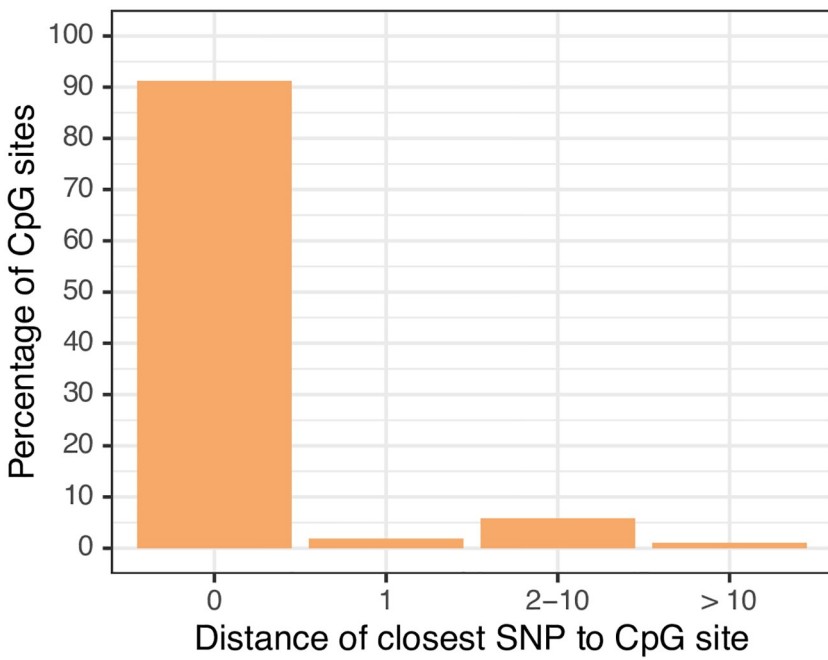

**Fig 5. Positions of known SNPs in probe sequences of correlated probes.** 1,507 of the 1,513 significantly correlated probes have known SNPs in their probe sequence. The vast majority of these (> 90%) map to the CpG site itself.

## Limited evidence for converging associations between DNA methylation and obesity from whole blood and sperm

We next investigated whether obesity was associated with DNA methylation in sperm or blood. At the 697,384 sites that passed quality control in the combined replication group, including lean and obese males, we used linear regression of DNA methylation on obesity status, controlling for estimated blood cell types in the blood dataset. No probes passed array-wide significance ($P < 9 \times 10^{-8}$) in blood or sperm (S13 Table). Given our small sample size, we leveraged published data from a larger EWAS of BMI in whole blood [1]; see Materials and methods). First, we tested whether the 187 replicated array-wide significant probes ($P < 1.0 \times 10^{-7}$) reported by Wahl and colleagues, which were also present in our data, were enriched in lower-ranked P values in our data, and secondly, we compared effect sizes at these 187 probes between our samples and the published data. To make both analyses comparable we treated BMI as a continuous measure for these comparisons—as Wahl and colleagues had done in the original epigenome-wide association study. Both analyses confirmed enrichments of the reported associations in blood but not sperm: lower-ranked P values were enriched in blood ($P < 1.3 \times 10^{-23}$, Wilcoxon rank sum test) but not sperm ($P = 0.06$, Fig 6A) and similarly, the reported effects at the 187 probes were correlated significantly with effects observed in our blood data ($\rho = 0.72$, $P < 1.0 \times 10^{-50}$, Spearman's rank correlation, Fig 6B) but not in sperm ($\rho = 0.13$, $P = 0.11$, Fig 6C). This indicates that the associations identified by Wahl and colleagues do not generalize to sperm. Next, to maximise power within our own sample, we ran a linear mixed effects model across the discovery and replication datasets, using the 692,265 probes that survived quality control in both datasets. DNA methylation was regressed onto tissue (blood versus sperm), age, batch and obesity status, while controlling for interindividual variation with a random effect (S13 Table). This analysis found that methylation at one

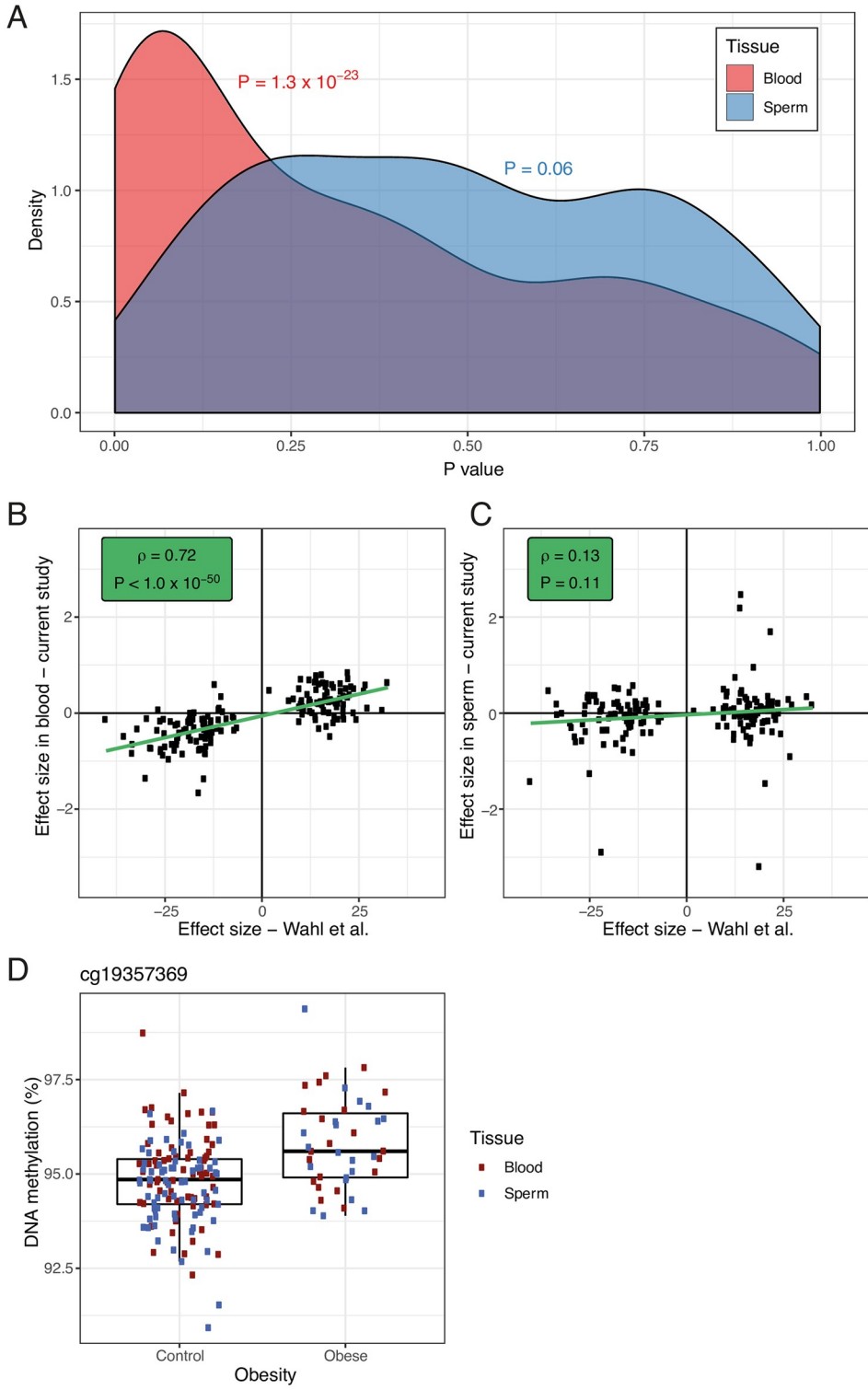

**Fig 6. Obesity associated DNA methylation patterns in whole blood and sperm.** Out of all replicated CpG sites reported to be associated with BMI by Wahl et al. ($P < 1.0 \times 10^{-7}$), 187 were also present in our replication group of lean and obese men. We regressed BMI onto DNA methylation in each tissue, controlling for estimated blood cell types in the blood analysis to match the analysis used by Wahl and colleagues. (**A**) Lower-ranked P values were found to be enriched amongst these 187 sites in blood ($P < 1.3 \times 10^{-23}$, Fisher's exact test) but not sperm ($P = 0.06$). (**B**)

Effect sizes at the 187 probes were significantly correlated between our blood data and the summary statistics published by Wahl and colleagues (ρ = 0.72, $P < 1.0 \times 10^{-50}$, Spearman's rank correlation). (**C**) No such correlation was observed for our sperm data (ρ = 0.13, $P$ = 0.11). (**D**) In a linear mixed effects model across the discovery and replication datasets, DNA methylation was regressed onto tissue (blood versus sperm), age, batch and obesity status, while controlling for interindividual variation. This analysis identified significant hypermethylation at one CpG site, cg19357369 (chr4:2429884), in obese compared to lean men across the two tissues (2% higher DNA methylation, $P = 8.95 \times 10^{-8}$).

CpG site, cg19357369 (chr4:2429884), was significantly increased in obese men in sperm and blood (2% higher DNA methylation, $P = 8.95 \times 10^{-8}$, Fig 6D). Finally, we compared our results with those of a previous study, which identified associations between paternal weight and off-spring DNA methylation in a sample of 429 father-mother-child triads [29]. Out of the nine probes at which Noor and colleagues found an association between cord blood DNA methylation and paternal periconceptional BMI, only one showed a nominally significant association in consistent effect direction in the sperm obesity EWAS ($P$ = 0.028, DNA methylation difference = 6%) and the combined mixed effects model in blood and sperm ($P$ = 0.01, DNA methylation difference = 4%). While the association in our data is weak and the probability of observing a false positive association across 18 tests (nine probes, two models) is almost 40%, the fact that the association of this probe is observed across both models and in consistent effect direction is encouraging and warrants further investigation.

## No association between obesity or metabolic traits and epigenetic age acceleration

Because obesity is associated with a higher risk for multiple age-related diseases, it has been suggested that this might occur via inducing accelerated cellular ageing [30]. Several studies used DNA methylation age acceleration—the discrepancy between a person's chronological age and their age predicted based on DNA methylation profiles—to investigate an association between obesity and accelerated ageing [30, 31], leading to inconsistent results. However, a recent meta-analysis [32] showed a small positive association between DNA methylation age acceleration in whole blood and BMI across seven studies. To test this association in our data, we derived three different estimates of DNA methylation age. In line with previous reports, we confirmed that the DNA methylation age estimator developed by Horvath [4] correlated significantly with chronological age when derived in whole blood ($r$ = 0.74, $P = 2.55 \times 10^{-9}$, Pearson's product moment correlation), but not in sperm ($r$ = 0.26, $P$ = 0.07, S7 Fig). This is likely because the Horvath DNA methylation was developed using only 45 samples of semen in a total of 7,844 samples (0.6%) of different tissue samples, including 4,180 blood-derived samples (53%) [4]. However, age could more accurately be predicted from sperm DNA methylation using the model recently developed by Jenkins and colleagues [33], which was specifically trained on sperm samples ($r$ = 0.68, $P = 1.78 \times 10^{-7}$, S7 Fig). The PhenoAge estimator [34], a biomarker of biological rather than chronological ageing, which has been shown to predict age-related traits and morbidity, was significantly correlated with chronological age in blood ($r$ = 0.73, P = $5.18 \times 10^{-9}$) but not sperm (r = 0.26, P = 0.08, S7 Fig).

We regressed DNA methylation age acceleration from the three models in blood and sperm onto five weight-related or metabolic traits: BMI, obesity (being in the obese/overweight group), waist circumference, insulin resistance (HOMA-IR) and fasting insulin. None of these 25 linear regressions identified significant associations between accelerated DNA methylation age and the five traits (P > 0.05 for all tests, S8 Fig, S14 Table).

## Obesity does not significantly influence the covariation of DNA methylation between sperm and blood

To investigate whether the covariation of DNA methylation was significantly altered in obesity, we ran an interaction model that regressed DNA methylation in blood onto DNA methylation in sperm, obesity status and their interaction effect, while covarying for experimental batch and age (see Materials and methods). We identified 98 CpG sites with a statistically significant interaction between obesity and the association of blood and sperm DNA methylation ($P < 9 \times 10^{-8}$). Interactions at the vast majority of these CpG sites (96) were driven by individual outliers in the obesity group; the remaining two sites appear to be driven by outliers in the lean group and a batch effect (S9 Fig). We therefore conclude that we were not able to identify credible altered DNA methylation covariation patterns between blood and sperm that may have arisen as part of a gene-environment interaction.

## Discussion

In this study, we characterized the sperm methylome in relation to blood and other somatic tissues, investigated covariation between DNA methylation in sperm and whole blood and analysed DNA methylation patterns associated with obesity. We conclude that the DNA methylation profiles of sperm and blood are highly distinct, and that there is little evidence of DNA methylation covariation between the two tissues, beyond genetic and technical effects.

In line with previous, smaller-scale studies, we showed that the sperm DNA methylome is highly polarised compared to that of blood, with both low (DNA methylation < 20%) and high (DNA methylation > 80%) levels of DNA methylation more frequently observed in sperm than in blood [20]. In contrast to previous research, however, we found that the sperm DNA methylome is overall slightly hypermethylated compared to that of blood [20, 35, 36]. This finding is potentially influenced by the fact that the previous generations of DNA methylation arrays (the 450K array) included a higher proportion of CpG islands, which are relatively hypomethylated in spermatozoa [20, 37].

We identified significant differences in DNA methylation levels at the majority of assayed CpG sites when comparing whole blood to sperm. Additionally, in our comparison of the spermatozoal DNA methylome to that of almost 6,000 somatic tissue samples, we showed that gene ontology terms enriched amongst hypermethylated CpG sites in sperm pointed repeatedly to transcriptional regulation. This is an intriguing finding considering that recent research has shown that high overall levels of transcription during spermatogenesis facilitate transcription-coupled DNA repair mechanisms through so-called "transcriptional scanning" [38]. Given that transcriptional regulation is an essential process for all cell-types, it is striking to observe sperm-specific DNA methylation patterns enriched in these processes. It could suggest that DNA methylation is involved in widespread transcriptional downregulation as cells progress from an active transcriptional stage during spermatogenesis to a more transcriptionally repressed stage in mature sperm.

About 1% of variable sites in whole blood and sperm showed a significant correlation of DNA methylation between the whole blood and sperm. This is slightly lower than what has been reported for comparisons of DNA methylation between whole brain and peripheral tissues [39]. Furthermore, at the vast majority of correlated CpG sites, the correlation appeared to be driven by underlying genetic variation resulting in characteristic bi- and trimodally clustered distributions of DNA methylation. In most of these cases, known SNPs were identified in the CpG site itself or in the single base extension. This finding is further supported by the observed enrichment of mQTLs [28] and CorSIVs [26] amongst correlated sites. Thus, whilst we lack specific genotyping information on individual participants in this study, our findings

strongly suggest genetic variation as the underlying cause of DNA methylation covariation between blood and sperm. This is despite the fact that we employed stringent filtering of probes in close proximity to SNPs from previously published lists [37, 40, 41], which suggests a need to update existing reference lists.

We also identified a small number of CpG sites where DNA methylation was negatively correlated between blood and sperm, and sites where DNA methylation exhibited a trimodal distribution pattern in one tissue only. It would be of interest to investigate further whether pathophysiological traits are associated with an increase in DNA methylation in one tissue and a decrease in the other. In particular, whether germ cell or leukocyte specific transcription factors are responsible for the discordant yet correlated DNA methylation distribution patterns across blood and sperm.

The small number of sites (6 out of 1,513) where no obvious genetic driver of methylation variability was identified are likely too few to be of value in studies where blood is needed as a surrogate tissue for sperm. The results of this study are generally in line with similar studies of DNA methylation covariation, such as between whole blood and various brain regions [15], albeit more extreme. They emphasize the importance of using disease-relevant tissues in epigenomic investigations. These findings do not however, generally preclude the use of readily accessible tissues such as blood or saliva for identifying DNA methylation biomarkers of conditions relating to germ cell function, such as subfertility. For example, if a robust DNA methylation profile of subfertility is identified in blood, this could be a helpful test in fertility evaluations without necessarily reflecting the epigenetic profile of spermatozoa.

This study identified one CpG site, cg19357369, as hypermethylated in sperm and blood from obese versus lean males. The finding should be interpreted with caution as it requires replication and just passed the array-wide multiple testing threshold—which was not corrected for the different aspects pertaining to sperm DNA methylation across the study (comparison with blood, correlation with blood, interaction, single-tissue EWAS, multi-tissue EWAS). The effect size was also comparatively small (2% higher DNA methylation in the obese group). cg19357369 is found upstream of the lncRNA *RP11-503N18*, which has yet to be characterised in terms of biological function [42]. However, previous research has shown that DNA methylation at cg19357369 is significantly altered during human fetal brain development [43]. Although cg19357369 has previously been identified as differentially methylated in hepatic tissue from obese compared to lean males [42], it has not previously been identified in EWASs of obesity or BMI when only blood samples have been analysed. If shown to be replicable, it could point towards the possibility of an obesity associated signature of spermatozoa.

Overall, we found that differentially methylated CpG sites associated with BMI in a large-scale EWAS in blood were not evident in sperm. Therefore, our current understanding of epigenetic associations of weight-associated phenotypes, which stems almost exclusively from studies of whole blood, is unlikely to give us functional insights into how these may be passed to offspring. Furthermore, in contrast to some previous reports, we did not identify any significant associations between obesity or metabolic traits and accelerated epigenetic ageing in blood or sperm.

There are limitations to our study. First, it constitutes an observational, cross-sectional study and we are therefore unable to comment on the causality behind observed associations between obesity and spermatozoal DNA methylation. The limited sample size of the obesity group (n = 22) reduced our ability to detect any modest association between obesity and DNA methylation covariation between sperm and whole blood. The obesity group included a proportion of overweight males (BMI 25–30 kg/m$^2$), which potentially diluted our results. Further, while we used the most comprehensive DNA methylation array currently available, the MethylationEPIC array is still biased towards certain parts of the genome (most notably

enhancer regions, RefSeq genes and CpG islands) and does not give a complete picture of genome-wide CpG methylation [44]. Lastly, although we were able to speculate as to the effects of genetic variants in CpG sites influencing our results, given trimodal methylation patterns and the presence of known SNPs in the CpG site, we did not have the actual genetic sequence of our subjects to verify this directly.

The study has several strengths. It constitutes the largest unbiased analysis of DNA methylation in matched human sperm and blood samples performed to date, and is one of the largest studies of spermatozoal DNA methylation in healthy males of proven fertility. In contrast to several previous analyses of DNA methylation in human spermatozoa [45–47], our study includes a replication group, increasing the robustness of our findings. Crucially, our analyses include the use of large existing datasets; blood-sperm correlated CpG sites were interrogated for overlap with previously identified mQTLs in whole blood [28], as well as with a list of recently reported CorSIVs [26]. We used findings from one of the largest studies of obesity-associated DNA methylation in blood performed to date [1] to analyse whether obesity-associated DNA methylation observed in blood was also reflected in spermatozoa. Lastly, we used recently developed DNA methylation analysis pipelines for large DNA methylation datasets [48] to identify sperm-specific DNA methylation signatures by comparing spermatozoal DNA methylation data to that of almost 6,000 somatic tissue samples available on GEO [21]. Together, these analyses allowed us to interrogate the spermatozoal DNA methylome in novel ways and provide highly suggestive evidence for why spermatozoal DNA methylation as a mechanism for intergenerational effects of obesity in humans is unlikely.

Recent research supports our conclusion that paternal BMI is unlikely to influence his offspring via DNA methylation. For example, a large-scale meta-analysis comprising almost 7,000 offspring found little evidence of an association between prenatal paternal BMI and offspring blood DNA methylation at birth or in childhood [49]. More research is warranted to help understand whether other epigenetic mechanisms, such as small RNA species, may be more influential in mediating effects of paternal obesity on offspring health, such as has been shown in non-human mammals [50, 51]. It would also be of interest to investigate the association between paternal traits other than BMI, such as smoking and ageing, and spermatozoal DNA methylation in an unbiased, genome-wide manner [52].

Our data suggests that compared with a wide range of somatic tissues, human sperm displays a unique DNA methylation profile, particularly in pathways relating to transcriptional regulation. We show that DNA methylation levels in human blood and sperm are only correlated at a minority of CpG sites and that at such sites, DNA methylation covariation is most likely due to genetic effects. The use of peripheral blood as a surrogate tissue for human spermatozoa is therefore inadvisable. Obesity does not generally influence spermatozoal DNA methylation, nor the covariation of DNA methylation between blood and sperm. Further, obesity-associated CpG sites identified in peripheral blood do not show enrichment in spermatozoa from obese individuals. Taken together, our findings suggest that if there are inter- and transgenerational effects of human obesity, they are unlikely to be mediated by changes in spermatozoal DNA methylation.

## Materials and methods

### Samples

Whole blood and semen samples were collected from participants recruited from University College London Hospital (UCLH) May 2016—March 2019. Participants were phenotyped with regards to BMI, waist circumference, systolic and diastolic blood pressure, blood lipids, fasting insulin and glucose levels and C-reactive protein (CRP). Two groups of participants

were included; lean (BMI $<25$kg/m$^2$) and overweight/obese (BMI $>26$kg/m$^2$). Phenotypic information about participants is detailed in S4 Table, which shows clear differences in metabolic variables between these groups. To determine BMI, participants were weighed wearing only light clothing and their height was measured by a trained researcher during the same research clinic visit as when their blood samples were taken, and within two weeks of providing a sperm sample. Participants provided information about their medical history and lifestyle via questionnaires, and were excluded if they suffered from significant medical conditions, took regular medications or smoked cigarettes. All participants were of proven fertility. Peripheral blood samples were centrifuged at 3000g for 15 minutes within one hour of venepuncture and the buffy coat was used for DNA extraction.

Semen samples were processed within one hour of sample production as per UCLH protocol and analysed for sperm concentration, motility and average progressive velocity using the Sperminator/Computer Assisted Sperm Analysis system (Pro-Creative Diagnostics, Staffordshire, UK). Semen sample parameters are detailed in S15 Table. All semen samples were within normal parameters according to World Health Organization criteria [53]. Samples underwent gradient centrifugation (45 and 90% PureSperm medium; PureSperm 100, Nidacon Laboratories, PS100-100) to select for motile spermatozoa as described elsewhere [54]. The processed samples were microscopically assessed for cell purity such that only samples with no visible cells other than spermatozoa were included in downstream analyses.

## Ethics approval and consent to participate

Ethical approval for the study was granted from the South East Coast—Surrey Research Ethics Committee on 28 September 2015 (REC reference number 15/LO/1437, IRAS project ID 164459). The study was also registered with the University College London Hospital Joint Research Office (Project ID 15/0548). All participants provided written, informed consent.

## DNA extraction

DNA from 200 μL buffy coat derived from whole blood was extracted using Qiagen QIAamp DNA Blood Mini Kit (Qiagen, Cat No. 51104) according to manufacturer's instructions [55]. DNA from the pellet of motile spermatozoa was extracted using a standard phenol-chloroform extraction method as described previously [56]. DNA extracted from whole blood and sperm was quality controlled using a Qubit 3.0 Fluorometer (Life Technologies, Cat No. Q33216). DNA was stored in -80˚C prior to bisulphite conversion.

## Methylomic profiling

DNA (500 ng) from each sample was sodium bisulphite-treated using the Zymo EZ 96 DNA methylation kit (Zymo Research, Cat No. D5004) according to the manufacturer's instructions. DNA methylation was quantified using the Illumina Infinium MethylationEPIC BeadChip [44] using an Illumina iScan System [57]. Samples were assigned a unique code for identification and randomized with regards to group and other variables to avoid batch effects, and processed in two batches. The Illumina Genome Studio software was used to extract the raw signal intensities of each probe (without background correction or normalization). Raw DNA methylation data is available for download from GEO (accession number GSE102538).

## Data pre-processing

Data analysis was performed in R version 3.6.2. DNA methylation data was processed and analysed using the *wateRmelon* package in R [58]. An initial outlier analysis was performed using

the outlyx() function in *wateRmelon* based on 1) the interquartile range of the first principal component and 2) the pcout algorithm [59] detecting outliers in high dimensional datasets, leading to the removal of 1 individual from the discovery group, 2 individuals from the obesity group and 3 Individuals from the lean replication group. The 59 non-CpG SNP probes on the array were used to confirm that the genotypes at these 59 probes were identical for the matched samples.

Prior to data analysis, 9,779 probes were removed from the discovery data because more than 5% samples displayed a detection P value > 0.05. Furthermore, 3,337 probes were removed because of having a bead count < 3. Probes containing SNPs in close proximity to the CpG site (within 10 base pairs) as well as potentially cross-reactive probes were filtered using annotated lists from three sources [37, 40, 41], leading to the removal of 149,105 CpG sites. The final discovery data set comprised 704,356 CpG sites. Data was normalized in the R package *wateRmelon* using the dasen() function as previously described [58]. The lean and obese replication groups were processed together experimentally and therefore jointly pre-processed and normalised using the same parameters as for the discovery dataset. A total of 697,442 probes survived quality control and filtering in the replication data. DNA methylation was analysed as beta values, which is the ratio of methylated probe intensity over the overall intensity and approximately equal to the percentage of methylated sites (% DNA methylation).

## Data analysis

**Characterization of DNA methylation in sperm.** CpG sites were assigned to chromosomes, locations, genes, and genomic regions using the Illumina manifest for the EPIC array (hg19 reference). CpG sites were classified as having either 'high' (median DNA methylation > 80%) or 'low' (median DNA methylation < 20%) DNA methylation. Enrichments of each genomic or CpG region amongst 'high' and 'low' methylation sites were calculated against the background (sites showing 20–80% median DNA methylation) using a Fisher's exact test.

**Annotation of imprinted genes/ imprinting control regions.** CpG sites were annotated to imprinted genes using the Illumina manifest for the EPIC array and the list of imprinted genes published in the Geneimprint database (http://www.geneimprint.com/site/genes-by-species). Enrichments of intermediate methylation levels were calculated as Fisher's exact tests of number of sites with 40–60% median DNA methylation levels annotated to imprinted genes against the array-wide background. For known human imprinting control regions (ICR) we used the locations reported by WAMIDEX [23], these were lifted to hg19 and overlapped with CpG locations using the R package *GenomicRanges* [60]. Enrichments for intermediately methylated (40–60% median DNA methylation) and unmethylated (median DNA methylation < 20%) sites were calculated as Fisher's exact tests.

**DNA methylation differences between blood and sperm.** Sites characterized by differences in DNA methylation between whole blood and sperm were identified by a paired t-test of matched samples. Comparison of the difference in DNA methylation levels between sperm and blood at different genomic regions was performed by calculating a paired t-test of median DNA methylation in sperm vs blood across all sites annotated to a specific genomic or CpG region.

**GEO analysis.** DNA methylation data for 6,288 samples was downloaded from the Gene Expression Omnibus (GEO) including 281 sperm samples and 5,971 somatic tissue samples from male donors, profiled using the 450K or EPIC arrays. Statistical analyses were performed using the *bigmelon* package in R and statistical tests were performed using *limma* [48, 61]. In the comparison of DNA methylation between sperm and tissue samples from males on GEO, a

linear model was fitted using the lmFit() function from the *limma* R package [61] across the 452,626 CpG sites that are present on both the EPIC and 450K arrays. The model regressed DNA methylation onto tissue (sperm vs not sperm) and included age and array type (450K or EPIC) as covariates. For sperm samples from GEO which lacked recorded age, the estimated age based on Jenkin's model was used instead. The data was not normalised because global large-scale differences between somatic tissues and sperm were expected, and because the high number of different types of samples included was expected to ameliorate issues around technical noise. We performed principal components analysis (PCA) of all samples from the 93 GEO datasets included in this analysis, to check for global effects of dataset or tissue of origin (S10 Fig). The gene ontology (GO) pathway analysis was performed using the gometh() function from the *missMethyl* R package [62], which removes ambiguously assigned probes from the enrichment analysis.

**Correlation between whole blood and sperm DNA methylation.** In order to minimise the effect single outliers would have on the correlation analysis, a subset of 'variable' probes was identified by calculating the DNA methylation difference between the 10th and 90th percentile across all samples, and selecting sites where this was at least 5% in both whole blood and sperm (n = 155,269 sites). This approach is similar to the one described by Hannon and colleagues previously [15]. Correlated CpG sites between sperm and blood were identified by Pearson's correlation test across all variable probes. In order to establish the matching null distribution, samples were permuted 100 times and correlations between DNA methylation in whole blood and sperm were recalculated across all variable sites. The density curve of these simulated correlations was added to the histograms of the empirical correlation coefficients to represent the null distribution (Fig 4). To investigate the clustering of DNA methylation patterns at significantly correlated CpG sites we used two separate methods: 1) kmeans method: a two dimensional outlier test was used by adapting the rosnerTest() function from the *EnvStats* R package [63] to exclude unimodal distributions. Next, k means clustering was applied for 2 and 3 clusters as implemented in the function pamk() of the R package *cluster* [64]. This function determines the best fitting number of clusters (two or three—corresponding to bi- and tri-modal methylation distributions). We manually checked and, if necessary, reassigned clusters which exhibited low between-cluster to within-cluster variance ratios (ratio < 2). 2) gaphunter: we applied the gaphunter() function from the Bioconductor package *minfi* [65] to blood and sperm DNA methylation values, identifying multimodal DNA methylation patterns in each tissue. This algorithm looks for consistent differences of > 5% DNA methylation, but only works on one-dimensional data, so had to be applied to each tissue separately.

**Annotation of SNPs and genetic enrichments.** To annotate SNPs to their location within probe sequences we used the Illumina EPIC hg38 manifest and dbSNP database build 151 in the *SNPlocs.Hsapiens.dbSNP151.GRCh38* R package. SNPs were mapped to probes using the *GenomicRanges* R package [60] and the distance to the CpG site of the closest SNP in the probe sequence was calculated for each of the 1,513 probes with significant correlations between sperm and blood. We downloaded the locations of the 9,226 correlated regions of systemic interindividual variation (CORSIV) in DNA methylation recently published by Gunasekara and colleagues [26]. These were overlapped with the locations of CpG sites using the hg38 manifest and the *GenomicRanges* R packages. Finally, we downloaded the list of cis methylation QTLs (mQTLs) in blood reported by McClay and colleagues [28]. These were identified using the 450K array, which meant we had to restrict this annotation to probes present on both the EPIC and 450K array. Enrichments for CORSIVs and mQTLs were calculated by Fisher's exact test against the background of non-correlated variable probes.

**Obesity and DNA methylation in blood and sperm.** Two models were used to investigate the association between obesity and DNA methylation in sperm and blood. First, DNA

methylation was regressed onto obesity status in the combined replication group, in blood and sperm separately. This analysis was controlled for estimated blood cell counts in blood. Secondly, a mixed effects model was run across both the discovery and replication groups using the lmer() function from the *lme4* package in R [66], regressing DNA methylation onto tissue (blood versus sperm), age, batch and obesity status, while controlling for interindividual variation with a random effect:

$$lmer(Methylation \sim Tissue + Age + Batch + Obesity + (1|ID))$$

Given our small sample size—especially in the obese group—we downloaded summary statistics from an EWAS of BMI in whole blood [1]. 187 of the replicated array-wide significant probes ($P < 1.0 \times 10^{-7}$) reported by Wahl and colleagues were also present in our dataset. To make our data comparable we treated BMI as a continuous measure for these comparisons, regressing BMI onto obesity status and controlling for estimated blood cell proportions in the blood analysis. We tested for an enrichment of lower ranked P values amongst the 187 previously reported probes in our analysis using a Wilcoxon rank sum test. Secondly, we looked at correlations of effect sizes reported by Wahl and colleagues and observed in our data across the 187 probes using Spearman's rank correlation to allow for study-specific biases.

**DNA methylation age estimates and age acceleration associations.** DNA methylation age was estimated on the discovery sample from both blood and sperm DNA methylation using Horvath's DNA methylation age estimator [4] as implemented in the *watermelon* R package. We additionally estimated DNA methylation age from sperm using the method described by Jenkins and colleagues [33] and from blood and sperm using the PhenoAge [34] estimator by uploading raw DNA methylation data to the DNA Methylation Age Calculator website (http://dnamage.genetics.ucla.edu). We additionally downloaded DNA methylation age acceleration scores for Horvath's estimator and PhenoAge from the website, using the residual based method, which accounts for estimated blood cell composition in the linear regression. We generated DNA methylation age acceleration scores for Jenkin's estimator by taking the residuals of the regression of Jenkin's DNA methylation age estimator onto chronological age.

DNA methylation age acceleration based on the three estimators was regressed onto five weight-related or metabolic traits across all samples from the discovery and replication groups: BMI, obesity (where all members of the obese/overweight group were defined as obese), waist circumference, insulin resistance (measured by the Homeostatic Model Assessment of Insulin Resistance (HOMA-IR)) and fasting insulin levels.

**Interaction between obesity, tissue and DNA methylation.** To detect and interaction between obesity and the association between blood and sperm DNA methylation we ran linear model regressing DNA methylation in blood onto DNA methylation in sperm, obesity status and their interaction effect, while covarying for experimental batch and age:

$$lm(Methylation_{Blood} \sim Methylation_{sperm} * Obesity + Age + Batch)$$

**Cell-type composition.** As whole blood represents a heterogenous tissue where the composition of leukocytes can introduce bias in the interpretation of DNA methylation analysis findings, blood cell type counts of monocytes, granulocytes, NK-cells, B cells, CD8+-T-cells, and CD4+-T-cells were estimated from the DNA methylation data using the method described by Houseman [67]. These estimates were included in all analyses that were run on the blood dataset alone as described above.

**Multiple testing correction.**    For agnostic analyses across the whole EPIC array (including those restricted to variable probes), the threshold $P < 9 \times 10^{-8}$ was applied as reported in recently published statistical guidelines for the EPIC array [68]. For the GEO analysis only the set of probes present on both the 450K and EPIC array were used. We applied Bonferroni correction across these 452,626 sites.

## Supporting information

**S1 Text. Replication.**
(DOCX)

**S1 Table. Enrichments of CpG region annotations across sites showing extreme methylation values in sperm.** Sites showing > 80% median DNA methylation were classified as "high", sites with median beta < 20% methylation as "low". Enrichments of each region amongst "high" and "low" methylation sites were calculated against the annotation of intermediately methylated sites (20–80% median DNA methylation) using a Fisher's exact test. *OR = odds ratio.*
(DOCX)

**S2 Table. Enrichments of genomic region annotations across sites showing extreme methylation values in sperm.** Sites showing > 80% median DNA methylation were classified as "high", sites with < 20% methylation as "low". Enrichments of each region amongst "high" and "low" methylation sites were calculated against the annotation of intermediately methylated sites (20–80% median DNA methylation) using a Fisher's exact test. *OR = odds ratio.*
(DOCX)

**S3 Table. Summary statistics for differences in DNA methylation between whole blood and sperm.** We used a paired t-test to identify DNA methylation differences between whole blood and sperm across all 704,356 probes passing quality control in the discovery dataset. Summary statistics are reported for all sites in the discovery dataset. Summary statistics from the replication groups are reported for sites that also passed quality control in our replication dataset. *IlmnID = Illumina CpG identifier, chr = chromosome, location = position on chromosome in hg19 reference, P = p-value in the discovery data, effect = effect size in the discovery data, P_rep = p-value in the lean replication group, effect_lean = effect size in the lean replication group, P_ob = p-value in the obese replication group, effect_ob = effect size in the obese replication group.*
(ZIP)

**S4 Table. Phenotype characteristics of participants included in the discovery, replication and obesity groups.** Reference ranges are derived from the UCLH Clinical Biochemistry Test Information sheet available from [70]. The reference range for HOMA-IR is derived from [71]. *SD = Standard Deviation, IQR = interquartile range, BMI = Body Mass Index, SBP = Systolic Blood Pressure, DBP = Diastolic Blood Pressure, HOMA-IR = Homeostatic Model Assessment of Insulin Resistance, CRP = C-Reactive Protein, HDL = High Density Lipoprotein, LDL = Low Density Lipoprotein.*
(DOCX)

**S5 Table. Blood and sperm DNA methylation difference by CpG region.** Using a paired t-test the DNA methylation difference between the median methylation in blood and sperm was calculated for each region. The DNA methylation difference is shown with respect to blood (a positive value indicating higher average DNA methylation in sperm).
(DOCX)

**S6 Table. Blood and sperm DNA methylation difference by genomic region.** Using a paired t-test the DNA methylation difference between the median methylation in blood and sperm was calculated for each region. The DNA methylation difference is shown with respect to blood (a positive value indicating higher average DNA methylation in sperm).
(DOCX)

**S7 Table. Details on non-sperm tissue samples in the GEO analysis.** The corresponding accession numbers are provided in S8 Table.
(DOCX)

**S8 Table. Accession numbers of all DNA methylation samples downloaded from GEO.**
(ZIP)

**S9 Table. Summary statistics for differences in DNA methylation between sperm and somatic tissue samples from GEO.** We compared DNA methylation in 371 sperm samples (including 90 samples from our dataset) to that of 5,917 somatic tissue samples from GEO using linear regression. This analysis was conducted across all 452,626 sites that are present on both the 450K and EPIC array. Summary statistics are reported for all sites. *IlmnID = Illumina CpG identifier, chr = chromosome, location = position on chromosome in hg19 reference, P = P value for difference between sperm and somatic cell DNA methylation, P_Bonferroni = Bonferroni-adjusted P value, effect = DNA methylation difference (beta)–negative values indicate lower DNA methylation in sperm compared to somatic tissues.*
(ZIP)

**S10 Table. Significantly enriched Gene ontology terms amongst CpG sites identified to be hypermethylated in sperm compared to somatic tissues.** GO analysis identified 272 pathways enriched amongst hypermethylated sites. Of note, 37 of these (14%) related to transcriptional regulation, while 67 (25%) were related to brain and neurological categories. *GO ID = Gene Ontology identifier, N = number of genes in the GO term, DE = number of genes that were differentially methylated, P.DE = P value for over-representation of the GO term, ONTOLOGY: BP = biological process, CC = cellular component, MF = molecular function.*
(ZIP)

**S11 Table. Significantly enriched Gene ontology terms amongst CpG sites identified to be hypomethylated in sperm compared to somatic tissues.** GO analysis identified 37 pathways enriched amongst hypomethylated sites. Eight of these pathways were related to sensory perception, specifically smell. *GO ID = Gene Ontology identifier, N = number of genes in the GO term, DE = number of genes that were differentially methylated, P.DE = P value for over-representation of the GO term, ONTOLOGY: BP = biological process, CC = cellular component, MF = molecular function.*
(ZIP)

**S12 Table. Summary statistics for correlation of DNA methylation between whole blood and sperm.** We used a Pearson's correlation test to identify CpG sites where DNA methylation was significantly correlated between whole blood and sperm This analysis was restricted to the 155,269 sites that showed met minimum variability criteria in both tissues (range of middle 80% > 5%). Summary statistics are reported for all sites in the discovery dataset. Summary statistics from the replication groups are reported for the sites that also passed quality control in our replication dataset. *IlmnID = Illumina CpG identifier, chr = chromosome, location = position on chromosome in hg19 reference, P = p-value in the discovery data, r = correlation coefficient in the discovery data, P_rep = p-value in the lean replication group, r_lean = correlation coefficient in the lean replication group, P_ob = p-value in the obese replication group,*

*r_ob = correlation coefficient in the obese replication group.*
(ZIP)

**S13 Table. Summary statistics for the association between DNA methylation and obesity in whole blood and sperm.** We regressed DNA methylation onto obesity status in our combined replication dataset, separately in whole blood and sperm, controlling for estimated blood cell type proportions in the blood analysis. We furthermore used a linear mixed effects model across the combined discovery and replication datasets, regressing DNA methylation onto obesity status, tissue type and batch while controlling for interindividual variation. Summary statistics for both analyses are reported—the LME results are restricted to sites available in both the discovery and replication datasets. *IlmnID = Illumina CpG identifier, chr = chromosome, location = position on chromosome in hg19 reference, P_blood = p-value in blood analysis, effect_blood = effect size in whole blood, P_sperm = p-value in sperm analysis, effect_sperm = effect size in sperm, P_mix = p-value in the mixed effects model, effect_mix = effect size in the mixed effects model. All effect sized are reported using the lean men as reference group.*
(ZIP)

**S14 Table. Associations between DNA methylation age acceleration and weight-related or metabolic traits.** DNA methylation age acceleration based on three different estimators was separately regressed onto five weight-related or metabolic traits: BMI, obesity (being in the obese/overweight group), waist circumference, insulin resistance (HOMA-IR) and fasting insulin. None of the 25 associations were significant ($P > 0.05$ for all tests).
(DOCX)

**S15 Table. Semen sample parameters for the discovery and replication groups (the lean replication group and the obesity group).** Semen sample parameters were measured using the Computer-Assisted Sperm Analysis (CASA)/Sperminator software (Pro-Creative Diagnostics, Staffordshire, UK). V = volume, C = concentration, SD = Standard Deviation, WHO = World Health Organization. Percentage A-D sperm refer to the proportion of spermatozoa in different motility grades where A = most motile and D = least motile. Reference ranges derived from [53].
(DOCX)

**S1 Fig. DNA methylation at CpG sites annotated to imprinted genes is enriched in intermediate levels of DNA methylation in blood, but not sperm.** DNA methylation annotated to known imprinted genes (Geneimprint database; http://www.geneimprint.com), showed a characteristic enrichment in sites with DNA methylation around 50% (+/- 10%) in whole blood—particularly, those genes known to be paternally imprinted ($P < 1.00 \times 10^{-50}$, Fisher's exact test), but also for maternally imprinted genes ($P = 9.19 \times 10^{-9}$) and a less pronounced enrichment in genes predicted to be imprinted paternally ($P = 0.01$) or maternally ($P = 0.04$). No such enrichment was observed in sperm ($P > 0.05$ for all four tests).
(TIFF)

**S2 Fig. Imprinting control regions are around 50% methylated in whole blood, while being nearly completely unmethylated in sperm.** Nearly all of the 169 CpG sites that are located in known imprinting control regions (ICRs) display intermediate DNA methylation levels in blood (57% of sites with median DNA methylation between 40 and 60%; $P < 1.00 \times 10^{-50}$, Fisher's exact test). Simultaneously, they appear to be completely unmethylated in sperm (94% of sites with median DNA methylation $< 20\%$, $P < 1.00 \times 10^{-50}$).
(TIFF)

**S3 Fig. Load of DNA methylation on first 20 principal components (PCs) in whole blood and sperm.** The first PC, which explained 51.41% of the total variance, clearly distinguishes between blood and sperm, making tissue/cell type the single biggest factor contributing to variation in DNA methylation across our samples.
(TIFF)

**S4 Fig. Differences observed between whole blood and sperm DNA methylation replicated across two replication datasets.** The effect sizes at the 441,764 significant probes from discovery, which were also present in the replication datasets, were highly correlated with those observed in the replication groups (lean group: $r$ = 98%, $P < 1.0 \times 10^{-50}$; obese group: $r$ = 0.99, $P < 1.0 \times 10^{-50}$).
(TIFF)

**S5 Fig. 365 of the 1,513 significantly correlated sites were driven by single outliers.** Shown is DNA methylation in whole blood and sperm from the discovery and replication datasets at (**A**) cg02474032 (chr16:87678659), (**B**) cg25554892 (chrX:70434406), and (**C**) cg07636088 (chr13: 31734946). We observed higher measured DNA methylation in the individual outlier at less than 2% of these 365 sites.
(TIFF)

**S6 Fig. Correlations which did not replicate were driven small numbers of individual outliers in the discovery data.** Of the 1,250 correlated probes also present in the replication data 173 (13%) show no evidence of correlation in the replication datasets ($r < 0.3$ in both datasets) (**A**) The majority of these sites (127 sites; 76%) were characterized by a single outlier in the discovery data, without any outliers in the replication datasets. One example is found at cg06819230 (chr16:67567158). (**B**) cg25253080 (chr10:14795564) represents the only incidence where a set of 5 outliers did not replicate in either replication group. (**C**) The biggest set of outliers which did not replicate contained 6 individuals, with no outliers in the replication data and was found at cg27045994 (chr8:284126). (**D**) The only trimodal distribution which did not replicate was observed at cg17118288 (chr1:218563763).
(TIFF)

**S7 Fig. DNA methylation age prediction in whole blood and sperm.** As reported previously, the DNA methylation age predictor by Horvath was significantly correlated with chronological age in whole blood but not in sperm. However, chronological age could be more accurately predicted from DNA methylation in sperm using the predictor more recently developed by Jenkins and colleagues. Like the Horvath methylation age estimator, the PhenoAge estimator showed stronger correlations with chronological age in blood than in sperm. Note that the lean and obese/overweight replication groups were combined into one "replication" group for these analyses.
(TIFF)

**S8 Fig. No significant associations between DNA methylation age acceleration and weight-related or metabolic traits.** Scatter plots of DNA methylation age acceleration based on three different estimators and four weight-related or metabolic traits (BMI, waist circumference, Homa-IR and fasting insulin) are shown in whole blood and sperm. Linear regressions were performed for each of these 20 comparisons: No significant associations were identified. Note that all discovery and replication groups were combined for this analysis.
(TIFF)

**S9 Fig. Statistically significant interaction effects were driven by outliers in either the obese or lean groups.** The majority of significant interactions between sperm and blood DNA

methylation and obesity were driven by single or very few outliers in the obesity group. (**A**) At cg23132872 (chr2:191882300), the correlation in obese individuals is driven by a single outlier. (**B**) At cg22086461 (chr8:77343728) the correlation in obese individuals is driven by two outliers. (**C**) At cg17166874 (chr7:155381422) the correlation in lean men is driven by four outliers in the discovery data and methylation at this site is also characterized by substantial batch effects. (**D**) At cg19778375 (chr12:297831) there appears to be a batch effect between the discovery and replication datasets that contributes to an observed correlation in the lean men from the discovery cohort, which is not present in the replication datasets. (TIFF)

**S10 Fig. PCA plots for all samples downloaded from GEO.** DNA methylation profiles were downloaded from 93 separate datasets on GEO. From a combined PCA, the loads on the first four principal components for each sample are shown here, coloured by dataset of origin and tissue (legend only shown for tissues). While individual datasets often only contain a single tissue of origin and therefore batch and tissue effects may overlap and there are clearly study specific effects, we also see similarities between samples of the same tissue type across datasets. (TIFF)

## Acknowledgments

We thank the technicians at UCL Genomics at the Great Ormond Street Institute of Child Health for processing of the Infinium MethylationEPIC Array, Anna Greco for her role in recruiting participants, and Dr Sara Hillman and Dr Rob Lowe for previous work and helpful discussions on DNA methylation studies of obesity.

## Author Contributions

**Conceptualization:** Fredrika Åsenius, Vardhman K. Rakyan, Sarah J. Marzi, David J. Williams.

**Data curation:** Sarah J. Marzi.

**Formal analysis:** Fredrika Åsenius, Tyler J. Gorrie-Stone, Sarah J. Marzi.

**Funding acquisition:** Vardhman K. Rakyan, David J. Williams.

**Investigation:** Fredrika Åsenius, Ama Brew, Yasmin Panchbhaya.

**Methodology:** Tyler J. Gorrie-Stone, Elizabeth Williamson, Sarah J. Marzi.

**Project administration:** Fredrika Åsenius, Sarah J. Marzi.

**Resources:** Fredrika Åsenius, Elizabeth Williamson, Vardhman K. Rakyan, Michelle L. Holland.

**Software:** Tyler J. Gorrie-Stone, Sarah J. Marzi.

**Supervision:** Vardhman K. Rakyan, Sarah J. Marzi, David J. Williams.

**Validation:** Sarah J. Marzi.

**Visualization:** Fredrika Åsenius, Sarah J. Marzi.

**Writing – original draft:** Fredrika Åsenius, Sarah J. Marzi.

**Writing – review & editing:** Fredrika Åsenius, Tyler J. Gorrie-Stone, Leonard C. Schalkwyk, Vardhman K. Rakyan, Michelle L. Holland, Sarah J. Marzi, David J. Williams.

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
