## [Decision Letter · Decision Letter 0]

9 Jun 2020

Dear Dr Marzi,

Thank you very much for submitting your Research Article entitled 'DNA methylation covariation in human whole blood and sperm: implications for studies of intergenerational epigenetic effects' to PLOS Genetics. Your manuscript was fully evaluated at the editorial level and by independent peer reviewers. The reviewers appreciated the attention to an important topic but identified some aspects of the manuscript that should be improved.

We therefore ask you to modify the manuscript according to the review recommendations before we can consider your manuscript for acceptance. Your revisions should address the specific points made by each reviewer.

[LINK]

Yours sincerely,

John M. Greally, D.Med., Ph.D.

Section Editor: Epigenetics

PLOS Genetics

Wendy Bickmore

Section Editor: Epigenetics

PLOS Genetics

Reviewer's Responses to Questions

**Comments to the Authors:**

Reviewer #1: This study aimed to characterize associations between obesity and DNA methylation of sperm and blood. Furthermore, the study compares 1) the DNA methylation landscape of sperm relative to blood in paired samples and among other non-paired tissues 2) the potential DNA methylation signature of obesity within sperm and blood. Genome-wide DNA methylation of paired blood-sperm samples was profiled with the EPIC array within a limited number of participants. However, the striking differences across tissues enabled the authors to characterize mostly discordant DNA methylation sites and ~1% DNA methylation sites which are concordant. Some limited evidence of an obesity methylation signature across tissues is presented. The study adds to our understanding of tissue specific signatures and the importance of considering genetics in EWAS. However, the limited number of samples and design of the study makes interpretation challenging. There are a several major concerns that need to be addressed below:

Major:

1. Abstract: The article refers to “effect” of obesity to DNA methylation. It should be revised to “association” as this study is cross-sectional and correlational in nature. Please rephrase any “effects” to associations for consistency.

2. Abstract: “Obesity only nominally influences sperm DNA methylation, making it an unlikely mediator of intergenerational effects of metabolic traits.” Given the sample size and design, this study has limited power to make such generalizable conclusions. Please rephrase to state that this was observed in this study only.

3. Author summary: The statement “although more research is needed, obesity in fathers may not affect the health of their children directly.” This is out of the scope of this paper, the paper only tested obesity DNA methylation associations no inferences can be made about many health endpoints which have been shown in rigorous epidemiological studies.

4. Results: The analyses of epigenetic clocks are surprising in the results as this is not mentioned as a hypothesis. Additionally, within the scope of the paper would be to test associations with obesity influencing epigenetic age but the authors don’t do this. I suggest leaving out or further elaborating on obesity and epigenetic aging. Particularly for the sperm clock.

5. Results: The trimodal distribution as a criterion of genetic control needs to be further explained. Are the authors suggesting that these measurements reflect a measurement of a SNP? (i.e. AA, AB, BB)? Please male sure to distinguish from meQTLs. As of now it is difficult to know if the authors suggest SNPs at the site/probe or meQTLs.

6. Results: Why was BMI chosen to be modeled as categorial, the authors need to elaborate between the modeling choice (continuous BMI vs categorical classification). One would expect to have greater power if modeling BMI as continuous. Additionally, it would be of high interest to see a scatter plot of cg19357369 DNA methylation and BMI as continuous.

7. In the methods no mention on how and when BMI was measured is presented. Need to elaborate, clinical visits? Self-report? and timing of ascertainment (i.e. during sample collection).

8. Discussion: a previous study reported associations between periconception obesity and child DNA methylation and birthweight (https://jamanetwork.com/journals/jamanetworkopen/fullarticle/2757997). It would be of high interest to do a reverse look-up of the 9 sites reported in that paper and findings in sperm and blood within this study.

Minor:

Abstract: please rephrased “Obesity only nominally influences sperm DNA methylation” given that the study was cross-section it would be more accurate to state “Obesity was nominally associated with sperm DNA methylation”.

Reviewer #2: This paper describes a study of DNA methylation in matched blood and sperm samples in relation to obesity (n=22 obese, n=68) not obese). The study conducts a number of interesting and relevant analyses, including: it characterises and compares the distribution of DNA methylation accross the genome, and particularly in imprinted regions, in both tissues; it calculates the correlation between chronological age and epigenetic age predicted by DNAm in both tissues; it compares the DNAm profile of sperm and that of somatic tissues using publicly available data; it calculates the correlation between blood and sperm at variable sites; it explores the likelihood of any blood-sperm correlations being driven by SNPs; it studies the association between obesity status and DNAm in blood and sperm in discovery and replication datasets; it explores enrichment for previously-identified BMI-associated CpGs in the blood and sperm EWAS results. The study concludes that the DNA methylation profiles of sperm and blood are highly distinct, and that there is little evidence of DNA methylation covariation between the two tissues, beyond genetic and technical effects.

The paper is well-written, clear and very interesting. It will be a valuable addition to the literature, which so far (perhaps partly attributable to publication bias) suggests paternal obesity and sperm methylation (and offspring methylation) are correlated. Most of these studies have looked at candidate genes and the field is in need of a genome-wide study with robust and comprehensive methods like this one.

I have only minor suggestions for how the manuscript might be improved.

Line 87-88 "[paternal factors] have the potential to negatively impact the development and physiology of a man's offspring, presumably via alterations to his spermatozoa." This sentence should be made more specific, or elaborated on to include other possibilities. Although changes to sperm are the most likely mechanisms responsible for direct prenatal effects, there are other possibilities, such as changes to the other constituents of semen and indirect effects via the mother. Of course, postnatal direct effects (eg through paternal behaviour) are also likely. For the same reason, I also suggest that in places where the authors state that their findings suggest that DNA methyaltion is an unlikely mechanism for intergenerational effects of obesity in humans (e.g. line 398), the wording be clarified (e.g. SPERM methylation, to distinguish from the possibility of paternal effects on maternal methylation or postnatal offspring methylation).

The authors have calculated the epigenetic age of sperm using the Horvath approach, and then calculated the correlation with, I think, chronological age (presumably this is what is meant by "works well"). The rationale for this analysis is not given but should be. Have the authors considered also looking at other epigenetic clocks, including phenoAge, and looking at the correlation between epigenetic age (acceleration) and obesity status?

The study compares DNAm in sperm (90 samples collected for this study, 281 publicly-available in GEO) to that of 5,917 somatic tissues from male participants from several other studies in GEO. They find that nearly 30% of the tested CpGs are differentially methylated after Bonferroni correction. They then carry out some functional enrichment analyses on the differentially methylated CpGs and find enrichment for gene transcription, neurological traits and sensory traits. They explain that the enrichment for neurological traits is possibly explained by the large proportion of brain and neuronal samples amongst the somatic tissues used. I would like to see how all samples cluster, for example in PCA plots, to be reassured that the major source of variation between sperm and somatic tissues is in fact tissue type and not technical variation introduced by the fact that the dfferent tissues will have been analysed by different labs using potentially different methods.

The authors may want to consider a recent preprint that supports the findings of their study: https://www.medrxiv.org/content/10.1101/2020.03.10.20020099v1

Reviewer #3: Plos Genetics

6/9/2020

DNA methylation covariation in whole blood and sperm: implications for studies of intergeneration epigenetic effects

Summary

Tissue specific epigenetic analyses are important and male semen samples are critically understudied. Authors conduct a tissue specific analysis of blood versus semen DNA methylation in adult males, first with discovery in 47 lean participants and then with replication testing in 21 lean participants. Authors specifically isolated active spermatozoa cell types in the semen samples and they adjusted for cell type proportions in the blood analysis. Widespread and large magnitude epigenetic differences were observed between these two tissues. Authors also conduct an analysis of the association between overweight or obesity status in 22 males versus lean status in both tissues. This weight analysis is likely underpowered and should be considered exploratory. In general, this manuscript is clear, the results are well quantitated, and the tissue specific research question is of high scientific value.

Critique

• Authors refer to study samples as cohorts, which implies longitudinal participant follow up. However, this is cross-sectional study. To avoid confusion, these should be described as samples.

• Authors use causal language at several points during the manuscript, which overstates the findings based on this study design. For example, in line 53 “We found that there was almost no effect of obesity on sperm”. Authors should be very careful to refer to these results as associations rather than causal effects.

• Epidemiology cross-sectional STROBE reporting guidelines require the study design (cross-sectional study) to be in the title and abstract.

• Authors frequently use the terms inheritance and intergenerational effects, however this is a cross-sectional study conducted in a single generation. Generational speculation should be restricted to the discussion and used much more cautiously. From the current title and phrasing throughout, this paper makes promises that are outside the bounds of the available data.

• The obese sample is misnamed. In fact, the mean BMI among this group is below 29, which reflects that this sample is overweight or obese.

• Recommend adding blood to Figure 1 as well for comparison.

• Authors are adequately powered for the tissue comparisons where effect sizes are expected to be large. In the overweight/obesity versus lean analysis, effect sizes are expected to be much smaller (can estimate from the CHARGE paper). The current study is likely vastly underpowered for this trait analysis. Authors should be more cautious in the interpretations that traits (such as BMI) are not associated with DNAm across these tissues. Perhaps a trait with larger expected effect sizes (such as smoking) with DNAm would be more appropriate benchmark?

• Probes with multimodal DNAm distributions can be identified agnostically from the data using gaphunter as implemented in the minfi package (PMID: 27980682). This is flexible by sample size and would allow the authors to have criteria beyond visual inspection for flagging these probes.

• Why were the samples preprocessed separately? It is confusing in the results when the total number of probes tested jumps around from 704k to 692k.

• Authors refer to the magnitude of association observed using beta. For example, “beta=0.2”. Since Illumina refers to beta values of methylation, there can often be confusion between beta values and beta coefficients. Recommend that the authors more fully interpret these coefficients in the text, for example, Overweight or obese participants had 20% higher DNA methylation than lean participants at X site.

• Can the authors please confirm that IDAT files are available at the GEO link and not just raw and processed methylation values?

• Is code available to conduct the analyses?

**Have all data underlying the figures and results presented in the manuscript been provided?**

Reviewer #1: Yes

Reviewer #2: Yes

Reviewer #3: None

PLOS authors have the option to publish the peer review history of their article (what does this mean?). If published, this will include your full peer review and any attached files.

Reviewer #1: No

Reviewer #2: No

Reviewer #3: No

---

## [Decision Letter · Decision Letter 1]

7 Aug 2020

Dear Dr Marzi,

We are pleased to inform you that your manuscript entitled "The DNA methylome of human sperm is distinct from blood with little evidence for tissue-consistent obesity associations" has been editorially accepted for publication in PLOS Genetics. Congratulations!

Yours sincerely,

John M. Greally, D.Med., Ph.D.

Section Editor: Epigenetics

PLOS Genetics

Wendy Bickmore

Section Editor: Epigenetics

PLOS Genetics

Comments from the reviewers (if applicable):

Reviewer's Responses to Questions

**Comments to the Authors:**

Reviewer #1: The authors have sufficiently addressed all major concerns. The additional analyses have also improved the overall quality and scope.

Reviewer #2: I would like to thank the authors for taking the time to write a very helpful Response to Reviewers and for addressing all my comments adequately.

**Have all data underlying the figures and results presented in the manuscript been provided?**

Reviewer #1: Yes

Reviewer #2: Yes

PLOS authors have the option to publish the peer review history of their article (what does this mean?). If published, this will include your full peer review and any attached files.

Reviewer #1: No

Reviewer #2: No

**Data Deposition**

http://datadryad.org/submit?journalID=pgenetics&manu=PGENETICS-D-20-00733R1

**Press Queries**

---

## [Editor Report · Acceptance letter]

6 Oct 2020

PGENETICS-D-20-00733R1 

The DNA methylome of human sperm is distinct from blood with little evidence for tissue-consistent obesity associations 

Dear Dr Marzi, 

We are pleased to inform you that your manuscript entitled "The DNA methylome of human sperm is distinct from blood with little evidence for tissue-consistent obesity associations" has been formally accepted for publication in PLOS Genetics! Your manuscript is now with our production department and you will be notified of the publication date in due course.

With kind regards,

Matt Lyles

PLOS Genetics

On behalf of:
